# Fractional Reasoning via Latent Steering Vectors Improves Inference Time Compute

## Abstract

Test-time compute has emerged as a powerful paradigm for improving the performance of large language models (LLMs), where generating multiple outputs or refining individual chains can significantly boost answer accuracy. However, existing methods like Best-of-N, majority voting, and self-reflection typically apply reasoning in a uniform way across inputs, overlooking the fact that different problems may require different levels of reasoning depth. In this work, we propose *Fractional Reasoning*, a training-free and model-agnostic framework that enables continuous control over reasoning intensity at inference time, going beyond the limitations of fixed instructional prompts. Our method operates by extracting the latent steering vector associated with deeper reasoning and reapplying it with a tunable scaling factor, allowing the model to tailor its reasoning process to the complexity of each input. This supports two key modes of test-time scaling: (1) improving output quality in breadth-based strategies (e.g., Best-of-N, majority voting), and (2) enhancing the correctness of individual reasoning chains in depth-based strategies (e.g., self-reflection). Experiments on GSM8K, MATH500, and GPQA demonstrate that Fractional Reasoning consistently improves performance across diverse reasoning tasks and models.

## 1 Introduction

Large Language Models (LLMs) have shown significant improvements across a variety of domains (OpenAI, 2023; Hurst et al., 2024; Anthropic, 2023; OpenAI, 2024; DeepSeek-AI et al., 2025). A key driver of their recent success is the paradigm of test-time compute: allocating additional computation at inference time to enhance reasoning ability (Qwen Team, 2024; Kimi Team et al., 2025; DeepSeek-AI et al., 2025). Typical strategies include generating multiple responses and selecting the best one (e.g., Best-of-N or majority vote), or iteratively refining answers through self-reflection or critique. These methods significantly improve performance without retraining, and are now central to the deployment of reasoning-focused LLMs.

However, current test-time compute strategies treat all problems uniformly. Each sample receives the same depth of reasoning (controlled by the same prompt), regardless of its difficulty or structure. In practice, reasoning needs are highly variable: simpler queries may be correctly answered with a single concise response, while harder problems benefit from deeper, more careful reasoning. Moreover, reasoning with under-, over-thinking or reflection can lead to degraded answers, or unnecessary computational costs (Chen et al., 2024; Pu et al., 2025). To fully realize the potential of test-time compute, LLMs need the ability to adapt their reasoning depth or level of reflection dynamically.

In this work, we introduce *Fractional Reasoning* (**FR**), a training-free and model-agnostic framework for improving test-time compute through adaptive reasoning control. The name reflects our core idea: rather than relying on fixed prompts that exert a uniform and non-adjustable influence, FR enables continuous control over reasoning intensity. Specifically, our method adjusts reasoning behavior by directly modifying the model's internal representations. We extract the latent shift induced by reasoning-promoting inputs (e.g., chain-of-thought or reflection prompts) and reapply this shift with a tunable scaling factor. This allows the model to modulate its reasoning depth at inference time, without altering the input text or requiring any fine-tuning. Our approach supports and enhances two key forms of test-time scaling: (1) Breadth-based scaling (e.g., Best-of-N, Majority vote): By tuning the level of reasoning in each generation, we increase the diversity and quality of outputs, leading to

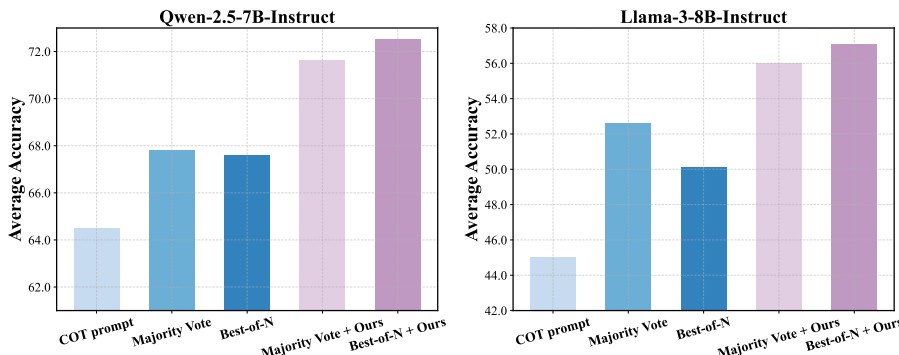

Figure 1: Averaged accuracy across MATH500, GSM8K, and GPQA. Blue bars represent standard test-time scaling methods, purple bars show these methods enhanced by our Fractional Reasoning.

higher success rates with fewer samples. (2) Depth-based scaling (e.g., self-reflection): By adjusting reflection strength, we enable fine-grained control over the level of reflection—how often and how strongly the model critiques and revises its output.

We evaluate our approach across multiple reasoning benchmarks using state-of-the-art open-source models. Across all settings, our method improves accuracy over standard prompting and offers enhanced flexibility in balancing under- and over-reasoning. We summarize our results in Figure 1. Furthermore, we show that our fractional reasoning framework generalizes to stronger reasoning-tuned models and scales robustly with the number of generations. Together, these contributions establish fractional reasoning as a general and interpretable paradigm for improving inference-time LLM scaling. We summarize our contributions as follows:

- We propose Fractional Reasoning, a general and training-free framework for adaptive reasoning control that enhances test-time compute by enabling fine-grained control of reasoning intensity.

- We present practical methods to extract and apply the effects of instructional prompts with tunable strength, requiring no fine-tuning or additional training.

- We demonstrate the effectiveness of our approach across multiple models and benchmarks (GSM8K, MATH500, GPQA), showing consistent improvements in both breadth-based (e.g., majority voting) and depth-based (e.g., reflection) test-time scaling strategies.

## 2 FRACTIONAL REASONING FRAMEWORK

To improve test-time compute with adaptive control on reasoning depth, we present the fractional reasoning framework that quantitatively controls the strength of instructional prompts to steer language model behavior. First, we formalize this view theoretically. Our key insight is that reasoning instruction prompts (e.g. chain-of-thought or reflection) induce directional shifts in the latent representations of the model. Then, we introduce how to explicitly control the strength of the prompt by identifying and reapplying such shifts at inference time without modifying the input text or fine-tuning the model.

### 2.1 PROMPT AS LATENT STATE SHIFT

Most LLMs employ Transformer (Vaswani et al., 2017) as their backbone architecture, which process input sequences through stacked self-attention layers. Inspired by Liu et al. (2024a), we interpret adding an instructional prompt (e.g., "Think step by step") as inducing a shift in the model's latent representations.

Let $\mathbf{X}_{\text{query}}$ and $\mathbf{X}_{\text{prompt}}$ denote the token embeddings of a query and an instructional prompt, respectively. Define $\mathbf{X}_{\text{concat}} := \text{concat}[\mathbf{X}_{\text{prompt}}, \mathbf{X}_{\text{query}}]$ as the input when the prompt is prepended. The attention output, where queries are computed from $\mathbf{m}$, and keys and values from $\mathbf{n}$, is defined as $\mathbf{h}(\mathbf{m}, \mathbf{n}) := \text{Attn}(\mathbf{m}\mathbf{W}_q, \mathbf{n}\mathbf{W}_k, \mathbf{n}\mathbf{W}_v)$.

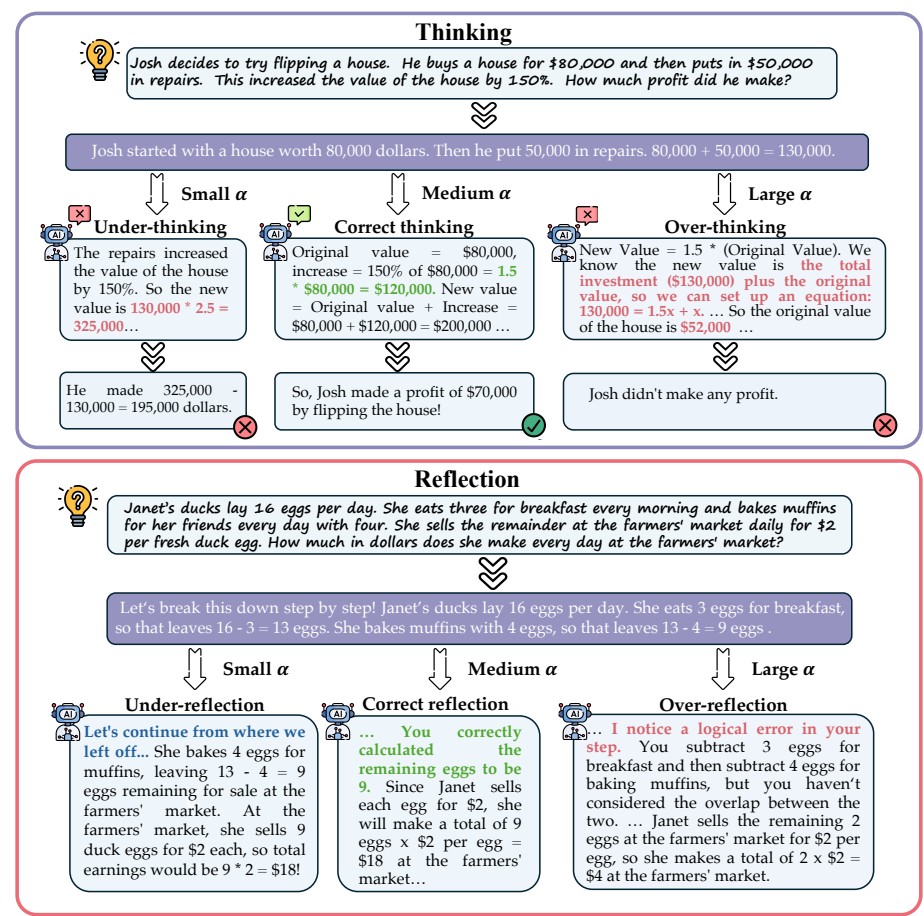

Figure 2: Example illustrating how model behavior changes with the scale of instructional strength $\alpha$ controlling the "fraction" of reasoning, applied to both Chain-of-Thought and Reflection prompting.

Under this notation, the output with prompt becomes:

$$\mathbf{h}(\mathbf{X}_{\text{query}}, \mathbf{X}_{\text{concat}}) = (1 - w)\mathbf{h}(\mathbf{X}_{\text{query}}, \mathbf{X}_{\text{prompt}}) + w\mathbf{h}(\mathbf{X}_{\text{query}}, \mathbf{X}_{\text{query}}), \quad (1)$$

where $w$ is a scalar determined by attention weights (proof and derivation in Appendix). This shows that adding an instructional prompt shifts the original self-attention output (i.e., $\mathbf{h}(\mathbf{X}_{\text{query}}, \mathbf{X}_{\text{query}})$) to additionally attending to the prompt (i.e., $\mathbf{h}(\mathbf{X}_{\text{query}}, \mathbf{X}_{\text{prompt}})$). However, the magnitude of this shift is fixed by the model's internal dynamics and not user-controllable.

## 2.2 FRACTIONAL REASONING FRAMEWORK

We propose a framework to explicitly control the strength of the shift at inference time. Let $\mathbf{h}_{\text{ori}} \in \mathbb{R}^{L \times d}$ denote the latent representations without prompt, and let $\mathbf{h}_{\text{steer}} \in \mathbb{R}^{L \times d}$ denote the latent steering vector capturing the prompt-induced shift in latent representation space, where $L$ is the number of layers and $d$ is the hidden dimensionality. We define the latent steering operation in our framework as:

$$\tilde{\mathbf{h}} := \text{Rescale}(\mathbf{h}_{\text{ori}} + \alpha \cdot \mathbf{h}_{\text{steer}}), \quad (2)$$

where $\alpha$ is a user-defined scalar controlling prompt strength, and $\text{Rescale}(\cdot)$ adjusts the norm of the steered latent states to minimize effect to subsequent modules and preserve stability across layers.

We construct the latent steering vector that summarizes the behavioral shift induced by a prompt. Drawing from Liu et al. (2024a), we compute this vector by generating contrastive example pairs $(\mathbf{X}^{\text{pos}}, \mathbf{X}^{\text{neg}})$, where the same query is prepended with a positive or negative prompt respectively, i.e., $\mathbf{X}^{\text{pos}} := \text{concat}[\mathbf{X}_{\text{pos\_prompt}}, \mathbf{X}_{\text{query}}]$ and $\mathbf{X}^{\text{neg}} := \text{concat}[\mathbf{X}_{\text{neg\_prompt}}, \mathbf{X}_{\text{query}}]$. A positive prompt

aligns with the original instructional prompt, while a negative prompt does the opposite. For example, for chain-of-thought prompting, a positive prompt might be "Solve the problem with step-by-step reasoning," and a negative one "Solve the problem by direct answering." We collect a set of $m$ contrastive pairs $\mathcal{X} = \{(\mathbf{X}_i^{\text{pos}}, \mathbf{X}_i^{\text{neg}})\}_{i=1}^m$ by adding the same positive and negative prompt to $m$ different queries, where $m$ is set based on the number of all queries. Then we feed each positive example and negative example in $\mathcal{X}$ into the LLM separately to obtain the latent representation. We extract the latent representations of the last token at each layer: $\mathbf{h}(\mathbf{X}) := \text{concat}\{\mathbf{h}^{(l)}(\mathbf{X}) | l \in [L]\} \in \mathbb{R}^{L \times d}$, where $\mathbf{h}^{(l)}(\mathbf{X}) \in \mathbb{R}^d$ is the latent states of the last token of example $\mathbf{X}$ at layer $l$. Intuitively, the latent steering vector should be closer to the representation of each positive example and further apart from that of negative ones. Given this, we define:

$$\mathbf{h}_{\text{steer}} := \arg\max_{\mathbf{h}} \frac{1}{m} \sum_{i=1}^m \left( \mathbf{h}^\top \left( \mathbf{h}(\mathbf{X}_i^{\text{pos}}) - \mathbf{h}(\mathbf{X}_i^{\text{neg}}) \right) \right)^2 \quad \text{s.t. } \mathbf{h}^\top \mathbf{h} = 1. \tag{3}$$

The solution of 3 is the first principal direction of the real valued set $\mathcal{Y} := \{\mathbf{h}(\mathbf{X}_i^{\text{pos}}) - \mathbf{h}(\mathbf{X}_i^{\text{neg}}) | i \in [m]\}$ (Liu et al., 2024a).

$\mathbf{h}_{\text{steer}}$ is a unit vector that captures the direction we would like to steer the LLM towards, We control the strength of the shift by explicitly scaling its length. Specifically, we add it to each token's latent states:

$$\hat{\mathbf{h}}_t := \mathbf{h}_t + \alpha \mathbf{h}_{\text{steer}}, \tag{4}$$

where $\mathbf{h}_t \in \mathbb{R}^{L \times d}$ is the latent state of the $t$-th token of query without instructional prompt. We instantiate the Rescale operation with norm preservation:

$$\tilde{\mathbf{h}}_t = \text{Rescale}(\hat{\mathbf{h}}_t) := \hat{\mathbf{h}}_t \cdot \frac{\|\mathbf{h}_t\|}{\|\hat{\mathbf{h}}_t\|}.$$

Our framework provides a unified and principled way to control prompt strength at inference time, supporting both interpretable behavior tuning and improved performance across reasoning tasks.

## 3 Experiments on Fractional Reasoning to Improve Test-time Compute

We evaluate our Fractional Reasoning framework as a tool for enhancing test-time compute through adaptive reasoning control. Our method enables quantitative adjustment of reasoning intensity at inference time, allowing the model to vary its behavior from concise direct answering to detailed multi-step reasoning and targeted reflection. We assess the impact of this adaptive control across multiple benchmarks and LLM families, demonstrating improvements in both breadth-based and depth-based test-time scaling strategies.

**Benchmarks and Models.** We evaluate the effectiveness of our method for improving test-time compute on GSM8K (Cobbe et al., 2021), MATH500 (Hendrycks et al.; Lightman et al., 2023), and GPQA (Rein et al., 2024), three widely used benchmarks that require multi-step reasoning. GSM8K focuses on grade-school math problems, MATH500 contains competition-style mathematical questions, and GPQA tests science-based reasoning from physics and natural science. These datasets span diverse reasoning types and problem difficulty, making them well-suited to test the adaptability of prompt strength. We use the test set for GSM8K and MATH500, and the diamond split for GPQA. Our main experiments use two competitive open-source instruction-tuned models: Qwen-2.5-7B-Instruct (Team, 2024) and LLaMA-3-8B-Instruct (Grattafiori et al., 2024), both of which demonstrate strong reasoning performance and provide access to latent state representations required by our method. All evaluations are conducted in a zero-shot setting.

**Baseline test-time compute methods.** Inference-time scaling methods in language models leverage additional computational resources during the inference phase to enhance performance by adaptively modifying the model's output distribution for a given prompt at test time. This process involves altering how responses are generated and processed to achieve more accurate or complex outputs compared to direct sampling from the model. We consider the following simple test-time compute methods.

- **Majority Vote.** Majority vote (self-consistency) generates multiple samples and chooses the most frequent answer as the final solution. Note that this method doesn't quite work for free-form generation problems such as LiveCodeBench. Hence, we don't present results for LiveCodeBench. For the reasoning model, we sample 100 samples for each query, while for non-reasoning models, we sample 256 samples.

- **Best of N.** Best of N samples N generations, and each generation is evaluated via a judge. We use a pretrained LLM `RLHFlow/Llama3.1-8B-PRM-Deepseek-Data` (Xiong et al., 2024) as a judge. The final answer is selected based on the judge's score.

**Evaluation Setup.** To construct the latent steering vector, the positive prompt is *"Solve the mathematics problem with step-by-step detailed reasoning"*, and the negative prompt is *"Solve the mathematics problem with direct answering"*. The vector is computed based on the positive and negative prompts appended with the queries. We perform fractional reasoning by applying this offset with varying scaling factors. For each query, we generate responses using 20 different prompt strengths uniformly sampled from a certain range (i.e., $\alpha$ values in Equation 2), creating a diverse ensemble of reasoning behaviors. An ablation study on the choice of $\alpha$ range is provided in Appendix A.3, showing that performance improves with wider ranges and stabilizes once sufficient diversity is introduced. For the baseline, we generate the same number of responses using the standard prompt without latent steering.

A key aspect of our evaluation is the connection between our motivation and our method. While our motivation suggests different problems require different $\alpha$ values, identifying the ideal $\alpha$ a priori is inherently difficult: it can vary significantly depending on the problem, the model's capability, and the nature of the reasoning that the specific task requires. Therefore, our framework is designed to solve this by first generating a rich candidate pool that creates a structured diversity across different reasoning intensities by sampling across a range of $\alpha$ values. This approach increases the likelihood of capturing the correct answer. We then use established test-time compute methods to effectively select the best $\alpha$ post-hoc. Specifically, we adopt a simple and widely-adopted setup: a verifier-free approach using *majority vote* across outputs with different $\alpha$ values, and a *reward-based* approach using best-of-$N$ selection guided by an external reward model. For the latter, to make a fair comparison to Best-of-N, we use `RLHFlow/Llama3.1-8B-PRM-Deepseek-Data` (Xiong et al., 2024) from Hugging Face to score each generation and select the one with the highest reward. This design directly addresses our motivation by using a candidate generation strategy that leverages different reasoning levels to improve the final output. Our proposed fractional reasoning brings structured diversity to majority vote and Best-of-N, allowing them to consistently recover stronger answers, much like a "Random Forest" benefits from many varied trees. Standard prompting is evaluated using the same protocols, either majority voting or selecting the highest reward from the same number of generations, to ensure fair comparisons. This avoids reliance on stochastic aggregation methods and allows us to directly isolate the effect of latent prompt strength variation.

**Results** Table 1 summarizes the results. Our method outperforms standard test-time compute methods on all benchmarks and models, demonstrating that our fractional reasoning framework can robustly enhance performance. The ability to vary prompt influence provides better coverage of the solution space, making conventional test-time compute methods more efficient. We also report results from frontier LLMs for reference.

## 4 FRACTIONAL REASONING FOR REFLECTION TO IMPROVE TEST-TIME COMPUTE

In this section, we explore how our Fractional Reasoning framework extends beyond chain-of-thought prompting to improve broader test-time compute strategies. Specifically, we focus on reflection prompting, which encourages post hoc reasoning and has been shown to enhance model performance by revisiting and revising initial responses (Pan et al., 2023). Reflection is a natural fit for fractional control, as different generations vary in their need for intervention: incorrect outputs benefit from stronger reflection, while excessive reflection on correct answers can lead to unnecessary changes or degraded quality. Our framework enables fine-grained adjustment of reflection strength, allowing the model to respond more appropriately to each case and avoid both under- and over-reflection.

Table 1: The performance of our proposed Fractional Reasoning (FR) and other common test-time scaling methods on different reasoning benchmarks is presented. The highest results are highlighted in **bold** and the second-best results are marked with underline. For some baselines, we use the results from their original reports or from Guan et al. (2025).

| Model | Datasets | | | Average |
| --- | --- | --- | --- | --- |
| | MATH500 | GSM8K | GPQA | |
| *Frontier LLMs* | | | | |
| GPT-4o* | 76.6 | 92.9 | 49.9 | 73.1 |
| Claude3.5-Sonnet* | 78.3 | 96.4 | 59.4 | 78.0 |
| GPT-o1-preview* | 85.5 | 94.9 | 73.3 | 84.6 |
| GPT-o1-mini* | 90.0 | 94.8 | 60.0 | 81.6 |
| *General Model: Llama-3-8B-Instruct* | | | | |
| Llama-3-8B-Instruct | 30.0 | 74.6 | 30.4 | 45.0 |
| Majority vote | 39.2 | 86.9 | 31.8 | 52.6 |
| Best-of-N | 36.6 | 79.1 | 34.7 | 50.1 |
| Majority vote + FR | **42.6** | 89.5 | 37.9 | 56.0 |
| Best-of-N + FR | 41.2 | **90.3** | **39.9** | **57.1** |
| *General Model: Qwen-2.5-7B-Instruct* | | | | |
| Qwen-2.5-7B-Instruct | 74.2 | 85.7 | 33.7 | 64.5 |
| Majority vote | 78.6 | 87.9 | 36.9 | 67.8 |
| Best-of-N | 77.2 | 91.2 | 34.3 | 67.6 |
| Majority vote + FR | **81.4** | 93.1 | 40.4 | 71.6 |
| Best-of-N + FR | 80.4 | **95.2** | **41.9** | **72.5** |

In reflection prompting, the input typically consists of multiple components: the reflection instruction, the problem description, and the initial generation. As a result, the input is generally much longer and semantically richer than in CoT prompting. To better accommodate this structure, we apply a minor modification to the instantiation of the latent steering operation (Equation 2). Instead of constructing contrastive examples, we directly use the latent states of the input with the reflection prompt as the latent steering vector. Specifically, let $\mathbf{X}_{\text{w/ prompt}} := \text{concat}[\mathbf{X}_{\text{reflection\_prompt}}, \mathbf{X}_{\text{query}}, \mathbf{X}_{\text{init\_generation}}]$ and $\mathbf{X}_{\text{w/o prompt}} := \text{concat}[\mathbf{X}_{\text{query}}, \mathbf{X}_{\text{init\_generation}}]$ denote the input sequences with and without the reflection prompt, respectively. We feed each into the LLM to obtain the latent states $\mathbf{h}(\mathbf{X}_{\text{w/ prompt}}) \in \mathbb{R}^{T_{\text{w/ prompt}} \times L \times d}$ and $\mathbf{h}(\mathbf{X}_{\text{w/o prompt}}) \in \mathbb{R}^{T_{\text{w/o prompt}} \times L \times d}$, where $T_{\text{w/ prompt}} = T_{\text{reflection\_prompt}} + T_{\text{w/o prompt}}$ denotes the total token length. To align the shapes, we pad $\mathbf{h}(\mathbf{X}_{\text{w/o prompt}})$ with zeros: $\mathbf{h}_{\text{w/o prompt pad}} := \text{Concat}[\mathbf{0}^{T_{\text{prompt}} \times L \times d}, \mathbf{h}(\mathbf{X}_{\text{w/o prompt}})]$. For the $t$-th token, let $\mathbf{h}_t := \mathbf{h}_{\text{pad}}[t, :, :] \in \mathbb{R}^{L \times d}$ be its original latent states, and $\mathbf{h}_{\text{steer}} := \mathbf{h}(\mathbf{X}_{\text{w/ prompt}})[t, :, :] \in \mathbb{R}^{L \times d}$ be the latent steering vector applied to it, we then compute the steered latent states in the same manner as previous: $\hat{\mathbf{h}}_t := \mathbf{h}_t + \alpha \mathbf{h}_{\text{steer}}$. We instantiate the Rescale operation as $\tilde{\mathbf{h}}_t = \text{Rescale}(\hat{\mathbf{h}}_t) := \frac{1}{1+\alpha} \hat{\mathbf{h}}_t$.

This form ensures norm stability and can be interpreted as a linear interpolation: let $\beta := \frac{\alpha}{1+\alpha}$, we have $\tilde{\mathbf{h}}_t = (1 - \beta)\mathbf{h}_t + \beta \mathbf{h}_{\text{prompt}}$. When $\beta = 1$, this reduces to standard prompting, and when $\beta = 0$ it approximates no prompt, aside from zero-padding, which we find to have negligible impact empirically. The resulting $\tilde{\mathbf{h}}_t$ is used as keys and values for subsequent attention computations. This adjustment improves effectiveness in longer contexts but does not alter the core framework.

The adaptation of steering vector's instantiation for reflection is a deliberate choice stemming from the fundamental differences between reflection and CoT reasoning. CoT represents a general behavior. We therefore compute a static, global steering vector from contrastive pairs, which can be reapplied to any new query to modulate its reasoning depth. Reflection, in contrast, is an instance-specific and context-specific intervention. The goal is to critique a specific initial generation for a specific query, meaning the intervention must depend on the immediate context. Our reflection method dynamically computes the steering vector from the latent shift induced by the reflection prompt for that specific instance. This distinction highlights the flexibility and generality of the Fractional Reasoning framework: it provides a general mechanism for control that is not limited to a single instantiation, allowing researchers to choose the most appropriate method for their task.

We use the same benchmarks and models as Section 3 . For evaluation setup, the model is provided with a query and an initial solution, which may be correct or incorrect. We use a verifier-free setup with majority voting.

The majority voting setting allows us to isolate the core benefit of our framework: by introducing diversity in reasoning strength, we shift the output distribution towards increasing the probability of sampling a correct answer.

Table 2 presents the results. Across all tasks and models, our method improves over baseline reflection prompting. Our framework controls levels of self-correction, which helps avoid the pitfalls of over-reflection and allows the model to better balance critique and preservation of valid reasoning.

Table 2: Reflection results. We report accuracy on GSM8K, MATH500, and GPQA. For model, "Qwen" refers to Qwen-2.5-7B-Instruct, and "Llama" to LLaMA-3-8B-Instruct. For settings, w/o Reflection denotes the accuracy of the initial solutions provided to the model before any reflection is applied; Standard Prompting applies a fixed reflection prompt; and our proposed Fractional Reasoning (FR) uses our framework with variable prompt strength. The best results are highlighted in **bold**.

| Model | Setting | Datasets | | | Average |
|---|---|---|---|---|---|
| | | MATH500 | GSM8K | GPQA | |
| | w/o Reflection | 23.6 | 75.8 | 31.8 | 43.7 |
| Qwen | Standard Prompting | 59.2 | 82.6 | 32.3 | 58.0 |
| Qwen | Fractional Reasoning | **61.4** | **84.9** | **35.4** | **60.6** |
| Llama | Standard Prompting | 30.4 | 78.9 | 28.3 | 45.9 |
| Llama | Fractional Reasoning | **31.8** | **80.1** | **32.3** | **48.1** |

**Toward Finer-Grained Sentence-Level Control.**    Beyond our previous experiments that apply a fixed $\alpha$ per query, finer-grained control finer-grained control at the sentence level allows the model to dynamically adjust $\alpha$ in response to emerging inconsistencies, enabling more precise and targeted corrections when needed.

We propose a strategy based on feedback signals, such as process reward model (PRM) score (Uesato et al., 2022; Lightman et al., 2023) or internal consistency metrics (Wang et al., 2022). The model begins with low $\alpha$ and increases it when the generation shows signs of inconsistency (e.g., low PRM score or low internal consistency), encouraging deeper reflection only when needed. This approach localizes correction efforts, improving precision. Figure 3 illustrates one such example on GSM8K with LLaMA-3-8B-Instruct. We convert the internal consistency metrics proposed in (Wang et al., 2022) into the reflection strength for each sentence, adjusting $\alpha$ dynamically throughout generation. In this case, the instance-level method fails to fix a reasoning error, whereas our sentence-level control successfully identifies and corrects the flawed step. This case study underscores the versatility of our method in supporting fine-grained, feedback-driven control and highlights a promising direction for future work: dynamic latent steering that adapts to evolving model states during generation.

## 5 ADDITIONAL ANALYSES

We further analyze our framework to understand its behavioral dynamics, generality across models, scalability with sampling budget, and potential for finer-grained control. Results throughout this section support the interpretability and flexibility of our latent steering framework.

### 5.1 FRACTIONAL REASONING CONTROLS MODEL BEHAVIOR

We analyze the effect of our fractional reasoning framework on model behavior through both quantitative and qualitative lenses. In each case, we demonstrate that varying the scaling parameter $\alpha$ leads to interpretable and controllable shifts in reasoning dynamics.

To assess how $\alpha$ influences reasoning verbosity, we measure the average length of model generations across different prompt strengths (i.e., $\alpha$ values) in Chain-of-Thought prompting. As shown in

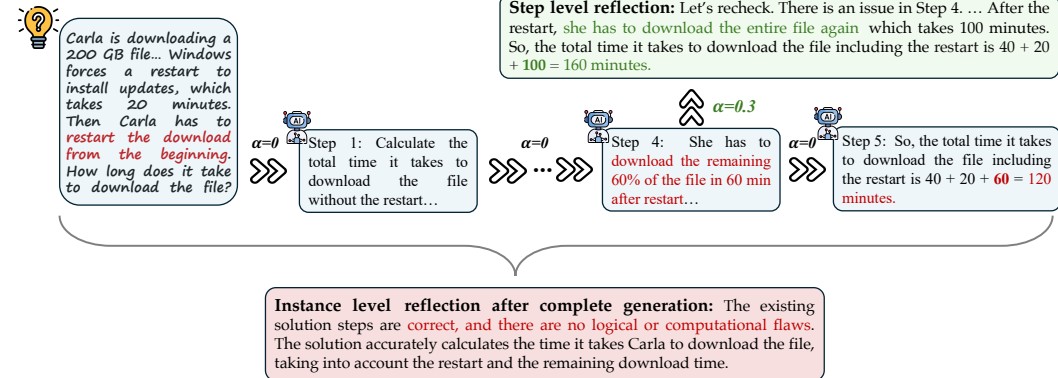

Figure 3: Sentence-level control dynamically adjusts reflection strength $\alpha$ at each generation step, enabling correction of errors missed by instance-level control.

Figure 4, increasing $\alpha$ leads to longer outputs, reflecting more detailed multi-step reasoning. This trend confirms that our framework steers model behavior in a predictable and continuous manner.

We further examine individual examples across different $\alpha$ values to qualitatively assess model behavior. Figure 2 presents representative outputs illustrating how the model's reasoning evolves with increasing prompt strength. At low $\alpha$, the model produces brief, shallow answers under CoT prompting and fails to exhibit reflective behavior in the reflection setting. At intermediate $\alpha$, it generates coherent multi-step reasoning for CoT and accurately critiques or improves prior steps in reflection. At high $\alpha$, we observe signs of over-thinking in CoT and over-reflection in the reflection setting, both of which can introduce unnecessary complexity and degrade final answer quality.

Table 3: Results for reasoning model: DeepSeek-R1-Distill-Qwen-7B. Fractional reasoning improves test-time scaling.

| Model | Datasets | | |
|---|---|---|---|
| | GSM8K | GPQA | MATH500 |
| DeepSeek-R1-Distill-Qwen-7B | 78.6 | 41.4 | 92.4 |
| Majority vote | 87.1 | 48.5 | 92.6 |
| Best-of-N | 88.7 | 41.1 | 92.4 |
| Majority vote + FR | 92.7 | **52.5** | **93.8** |
| Best-of-N + FR | **93.6** | 47.9 | 92.6 |

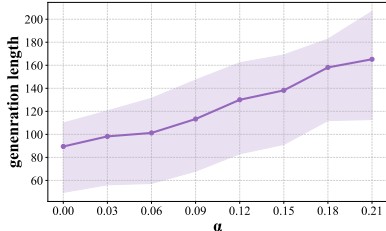

Figure 4: Mean generation length increases with larger $\alpha$ for COT prompting. Shaded area: 25% and 75% quartile across examples.

## 5.2 FRACTIONAL REASONING ON SCALING TEST-TIME COMPUTE FOR REASONING-TUNED MODELS

In addition to earlier experiments that on general-purpose instruction-tuned models, we also test our method on a reasoning-specialized model: DeepSeek-R1-Distill-Qwen-7B (DeepSeek-AI, 2025). We evaluate CoT prompting on GSM8K and GPQA datasets using the same split and setup as Section 3: 20 prompt strengths, selecting final answer based on majority vote or highest reward, and comparison against standard CoT prompting with repeated generation. As shown in Table 3, our method improves accuracy over the standard prompting baseline, showing that our fractional reasoning framework remains effective even when the underlying model is already optimized for reasoning. This result highlights the generality of our framework across both general and specialized LLMs.

## 5.3 EFFECT OF NUMBER OF GENERATIONS

We examine how test-time compute performance scales with the number of generated samples. Figure 5 reports accuracy as a function of the total number of generations on GSM8K and GPQA, following the setup in Section 3. We vary the number of prompt strengths (i.e., $\alpha$ values) in $\{2^0, 2^1, \ldots, 2^4\}$ and generate 5 responses per strength. For standard prompting baselines, we compare against both majority vote and best-of-$N$ selection using a reward model.

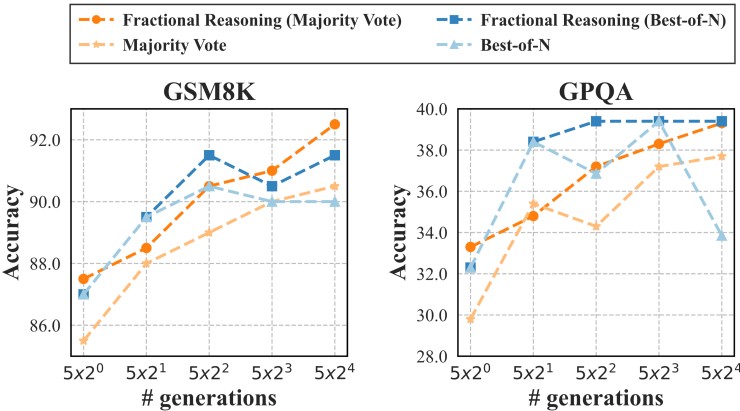

Figure 5: Accuracy on GSM8K and GPQA as a function of the number of generations.

Our method demonstrates consistent improvements as the number of generations increases. In contrast, the reward-based best-of-$N$ method does not scale as effectively, likely due to the increasing difficulty of reliably selecting the best response from a larger candidate pool. Compared to the majority vote baseline, our method shows higher accuracy across most sampling budgets, demonstrating that our fractional reasoning framework effectively alters the output distribution and increases the likelihood of sampling a correct response. These results validate that our method not only enhances reasoning performance but also scales robustly with increased generation. Moreover, our method consistently improves accuracy under different compute budget, offering a flexible and compute-efficient strategy for inference-time scaling.

## 5.4 DISCUSSION AND FUTURE DIRECTIONS

**Comparison with Textual Prompting Diversity.** We designed an experiment to examine whether it is effective to introduce diversity at the textual level, inspired by (Hu et al., 2025) . Specifically, we We created a "Diverse Prompts" baseline for Majority Vote by sampling from five different textual instructions: "Solve the problem with step-by-step detailed reasoning", "Solve the problem with direct answering", "Solve the problem with step-by-step detailed reasoning, think harder", "Solve the problem with step-by-step reasoning, think less", and "Solve the problem". On GSM8K using Llama-3-8B-Instruct, this Majority Vote + Diverse Prompts baseline achieved an accuracy of 85.2%, which underperformed our Majority Vote + Fractional Reasoning (89.5%). This result highlights the limitations of using simple prompt engineering to control reasoning intensity and strengthens the advantages of our more principled, latent-space steering framework.

**Extention to Multi-agent Collaboration Frameworks.** A promising future direction is to apply our Fractional Reasoning framework within complex multi-agent collaboration frameworks (Chen et al., 2023; Du et al., 2023). This could enable more sophisticated, heterogeneous teams by allowing for fine-grained control over the reasoning intensity of individual agents, effectively assigning them different roles.

**Locus of Intervention.** We note that our choice of intervention that applies the steering vector to the last-token representations concatenated across all layers is one of many possible instantiations. While a deep ablation on the optimal locus of intervention (e.g., specific layers, attention heads or MLP blocks) is an important and valuable research question, our main focus is to propose the general Fractional Reasoning framework rather than to identify a single optimal instantiation. We view a systematic study of the optimal intervention locus as a promising direction for future work.

**PRM choice for Best-of-N.** In our Best-of-N experiments, we use a strong, publicly available PRM to demonstrate our framework's effectiveness in this widely-used setting. While a deep analysis of sensitivity to different reward models is a valuable but distinct research direction, our framework is flexible and can be paired with any PRM.

## 6 RELATED WORK

**Inference-time scaling.** Test-time scaling has emerged as a powerful strategy for improving LLMs without requiring additional training. A prominent example is Chain-of-Thought (CoT) prompting (Wei et al., 2022) , which improves reasoning performance by guiding the model to decompose complex problems into intermediate steps. Another line of work focuses on evaluating and selecting high-quality responses. Outcome reward models (ORMs) and process reward models (PRMs) assess generated responses based on correctness or internal reasoning quality, enabling selective retention or reranking (Uesato et al., 2022; Lightman et al., 2023; Zhang et al., 2024; Luo et al., 2024). Complementary to reward models, self-consistency methods offer a verifier-free option that measure response agreement across samples (Wang et al., 2022). Another parallel line of work focuses on revision, where the model is prompted to reflect on and iteratively improve its own output, as in self-correction or reflection-based prompting (Madaan et al., 2023; Shinn et al.; Pan et al., 2023).

**Latent states control and representation editing.** Early work on activation-based control, such as Plug-and-Play (Dathathri et al.) introduced the idea of modifying model activations using attribute-specific classifiers to steer generation toward desired targets. Zou et al. (2023) proposed representation engineering, which uses steering vectors in the latent space of LLMs to enable controlled and interpretable generation. Subramani et al. (2022); Turner et al. (2023) showed that learned or predefined steering vectors can effectively shift model behavior. Li et al. (2023) demonstrated that manipulating attention head outputs can improve truthfulness in LLMs through inference-time intervention. More recently, Liu et al. (2024a) proposed In-Context Vectors (ICV), which extract latent vectors from demonstrations to steer internal states at inference time, enabling more controllable in-context learning. Wu et al. (2024) introduced Representation Finetuning (ReFT), a parameter-efficient approach that learns task-specific low-rank interventions over latent representations, often matching or exceeding fine-tuning performance with reduced overhead. Liu et al. (2024b) introduced latent shifting vectors for reducing hallucination of multimodal language models. Recently, Lin et al. explores using steering vectors to control thinking speed in reasoning models. While we also leverage latent steering, our novel, orthogonal contribution is the Fractional Reasoning framework focused on improving test-time scaling performance, which is generally applicable to distinct settings including breadth-based CoT and depth-based Reflection.

## 7 CONCLUSION

We present Fractional Reasoning, a training-free and model-agnostic framework for improving test-time compute through adaptive control of reasoning behavior in LLMs. By identifying and reapplying reasoning-induced latent shifts with a tunable scaling factor, our method enables continuous adjustment of both reasoning depth and reflection strength—tailoring inference-time behavior to the demands of each input. Experiments across multiple benchmarks and models show that Fractional Reasoning improves performance, stability, and sample efficiency under both breadth-based (e.g., best-of-n) and depth-based (e.g., self-reflection) scaling strategies. Our approach provides a general, interpretable mechanism for precise and efficient allocation of computational effort during inference.

## ETHICS STATEMENT

Our work uses only open-source models and publicly available benchmarks. No private or sensitive data are involved. While improved reasoning may be misused, our method does not alter model knowledge and we encourage deployment with proper safeguards. As our framework offers fine-grained control over a model's reasoning style, this capability could be misused by malicious actor. While our work is intended for beneficial applications, we believe addressing these broader societal impacts and potential risks of misuse provides a more complete and responsible discussion.

## REPRODUCIBILITY STATEMENT

All datasets (GSM8K, MATH500, GPQA), models (LLaMA-3-8B, Qwen-2.5-7B), and hyperparameters are described in the paper and appendix. Code is provided in the supplementary material to replicate results.

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

# A    TECHNICAL APPENDICES AND SUPPLEMENTARY MATERIAL

## A.1    DERIVATION OF PROMPT AS LATENT STATE SHIFT

In Section 2.1, we briefly show the effect of prompt on the latent states. Here we present a detailed derivation of Equation 1 . We follow the same notation as in the main text.

$$
\begin{aligned}
\mathbf{h}(\mathbf{X}_{\text{query}}, \mathbf{X}_{\text{concat}}) &:= \text{Attn}(\mathbf{X}_{\text{query}}\mathbf{W}_q, \mathbf{X}_{\text{concat}}\mathbf{W}_k, \mathbf{X}_{\text{concat}}\mathbf{W}_v) \\
&= \text{Attn}(\mathbf{X}_{\text{query}}\mathbf{W}_q, \text{concat}[\mathbf{X}_{\text{prompt}}, \mathbf{X}_{\text{query}}]\mathbf{W}_k, \text{concat}[\mathbf{X}_{\text{prompt}}, \mathbf{X}_{\text{query}}]\mathbf{W}_v) \\
&= \text{softmax}\left(\mathbf{X}_{\text{query}}\mathbf{W}_q\left(\text{concat}[\mathbf{X}_{\text{prompt}}\mathbf{W}_k, \mathbf{X}_{\text{query}}\mathbf{W}_k]\right)^\top\right)\begin{pmatrix}\mathbf{X}_{\text{prompt}} \\ \mathbf{X}_{\text{query}}\end{pmatrix}\mathbf{W}_v \\
&= (1-w)\,\text{softmax}\left(\mathbf{X}_{\text{query}}\mathbf{W}_q\left(\mathbf{X}_{\text{prompt}}\mathbf{W}_k\right)^\top\right)\mathbf{X}_{\text{prompt}}\mathbf{W}_v \\
&\quad + w\,\text{softmax}\left(\mathbf{X}_{\text{query}}\mathbf{W}_q\left(\mathbf{X}_{\text{query}}\mathbf{W}_k\right)^\top\right)\mathbf{X}_{\text{query}}\mathbf{W}_v \\
&= (1-w)\text{Attn}(\mathbf{X}_{\text{query}}\mathbf{W}_q, \mathbf{X}_{\text{prompt}}\mathbf{W}_k, \mathbf{X}_{\text{prompt}}\mathbf{W}_v) \\
&\quad + w\text{Attn}(\mathbf{X}_{\text{query}}\mathbf{W}_q, \mathbf{X}_{\text{query}}\mathbf{W}_k, \mathbf{X}_{\text{query}}\mathbf{W}_v) \\
&=: (1-w)\mathbf{h}(\mathbf{X}_{\text{query}}, \mathbf{X}_{\text{prompt}}) + w\mathbf{h}(\mathbf{X}_{\text{query}}, \mathbf{X}_{\text{query}})
\end{aligned}
$$

where

$$
w = \frac{\sum_i \exp\left(\mathbf{X}_{\text{query}}\mathbf{W}_q\left(\mathbf{X}_{\text{query}}\mathbf{W}_k\right)^\top\right)[i]}{\sum_i \exp\left(\mathbf{X}_{\text{query}}\mathbf{W}_q\left(\mathbf{X}_{\text{query}}\mathbf{W}_k\right)^\top\right)[i] + \sum_j \exp\left(\mathbf{X}_{\text{query}}\mathbf{W}_q\left(\mathbf{X}_{\text{prompt}}\mathbf{W}_k\right)^\top\right)[j]}
$$

## A.2    EXPERIMENT DETAILS

In this section, we provide additional details on the experimental settings that were not included in the main text.

### A.2.1    MODELS

The study evaluates a broad spectrum of models spanning various sizes and architectures to comprehensively assess the effectiveness of inference-time scaling methods. These models are grouped into non-reasoning and reasoning categories, based on their core capabilities and training objectives.

**Non-reasoning models**    Non-reasoning models refer to general-purpose LLMs primarily optimized for tasks such as text generation and dialogue, without specialized training for complex reasoning. Since our proposed method involves manipulating the models' latent states, we restrict our experiments to open-source models. The selected models include Qwen2.5-7B-Instruct Yang et al. (2024) and Llama-3-8B-Instruct Dubey et al. (2024). This selection ensures compatibility with our intervention approach while representing strong baseline performance.

**Reasoning models**    Reasoning models are specifically trained or designed to handle complex reasoning tasks, such as mathematical problem-solving and code generation, often through methods like reinforcement learning (RL). The selected model in our experiments inlude The selected models includes DeepSeek-R1-Distill-Qwen-7B due to the high-cost nature of inference-scaling methods.

### A.2.2    IMPLEMENTATION DETAILS

**Evaluation**    We evaluate our method on 3 different reasoning datasets, including math reasoning (GSM8k, MATH500) and general domain reasoning (GPQA). The final accuracy across these datasets and various models is reported.

**Experiments for depth of thinking**    We consider three baselines for comparison: the original model, Best-of-N, and majority vote. Here are the experiment settings for both baselines and Fractional Reasoning :

- **Original model.** For the original model, we use a temperature of 0.7, max_new_tokens set to 2048, and all other parameters at their default values. For reasoning models, we increase max_new_tokens to 8192 because thinking tokens require longer generations and adjust the sampling parameters to temperature = 0.6, top_k = 40, and top_p = 0.95. The number of generations is set to be 1.

- **Best-of-N.** We use the same generation hyperparameters as the original model, but set the number of generations to $N$. We employ a reward model from Hugging Face, `RLHFlow/Llama3.1-8B-PRM-Deepseek-Data`, to select the highest-scoring answer among the $N$ candidates. This reward model is fine-tuned from `meta-llama/Llama-3.1-8B-Instruct` on `RLHFlow/Deepseek-PRM-Data` for a single epoch. During implementation, we observed that formatting mismatches sometimes led to incorrect evaluations. To address this, we do not use explicit formatting prompts during inference; instead, we allow the LLM to complete the reasoning and call the LLM with the previously generated reasoning to obtain the final answer in a consistent format. The same model being evaluated is used for formatting.

- **Majority vote.** The generation hyperparameters match those of the original model, with the number of generations set to $N$ for comparability with Best-of-N. Instead of using a reward model, we first format each generated response and then use majority voting to select the most frequent answer as the final output.

- **Fractional reasoning.** We apply both slow and fast thinking modes for each question, controlled by a scaling factor $\alpha$ uniformly sampled from $(-0.15, 0.15)$. For each of the $N/5$ sampled $\alpha$ values, we generate 5 responses, totaling $N$ generations, to make a fair comparison with other test-time scaling methods. As with the other baselines, the LLM produces the final answer either by best-of-N or majority vote. To ensure fair comparison with Best-of-N and majority vote, all settings are kept the same except for the addition of latent steering via varying $\alpha$ values.

**Experiments for reflection** To evaluate reflection ability, the model is given a query and an initial solution, which may be correct or incorrect. The task of the model is to reflect on the initial response and correct any errors. The improvement in accuracy after reflection compared to the accuracy of the initial responses serves as a measure of reflection effectiveness. In our fractional reasoning framework, we apply varying reflection strengths by sampling a scaling factor $\alpha$ from $[0, 1]$. For each query and initial generation, we generate responses with three different reflection strength (i.e., $\alpha$ values) and select the final answer via majority vote. To ensure a fair comparison, the baseline prompt method also generates the exact same number of responses and selects its final output using the same voting strategy. During implementation, we found that answers were sometimes marked incorrect due to formatting discrepancies. To address this, we refrain from using explicit prompts to enforce answer formatting at inference time. Instead, we rely on the LLM to complete the reasoning process and produce final answers in a consistent format on its own.

### A.3 ABLATION STUDY

**Robustness to number of pairs** $m$**.** As we can observe from Table 4, Fractional Reasoning (FR) is robust across different $m$, which is the number of queries used to calculate the latent steering vector.

### A.4 EXAMPLES OF CONTROLLING THINKING DEPTH

Table 4: Performance with Majority Vote + Fractional Reasoning across different $m$.

| $m$ | GSM8k (Majority Vote + FR) | GPQA (Majority Vote + FR) |
|---|---|---|
| 128 | 89.3 | 36.9 |
| 256 | 89.5 | 37.9 |
| 512 | 89.6 | 37.4 |

**Robustness to $\alpha$ range.** We further conducted an ablation study on Llama-3-8B-Instruct to evaluate the sensitivity of Fractional Reasoning with majority vote to the sampling range of $\alpha$. As shown in

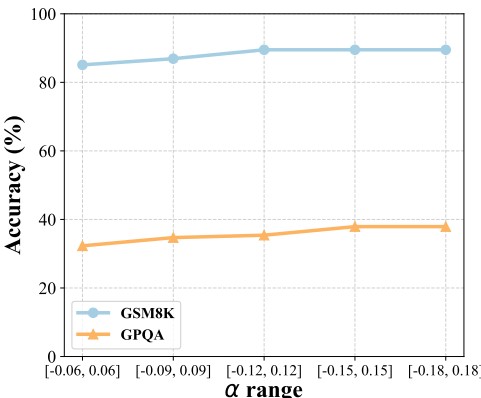

Figure 6: Ablation on $\alpha$ sampling range with Llama-3-8B-Instruct.

Figure 6, both GSM8K and GPQA benefit from moderate expansion of the range, with accuracy steadily improving until it saturates around $[-0.15, 0.15]$. Beyond this point, performance remains stable, indicating that our method is robust to the choice of range once sufficient diversity in reasoning strength is introduced.

## B  MORE EXAMPLES

---

**Example 1: Original generation (GSM8k; Llama-3-8b)**

*Query: Carla is downloading a 200 GB file. Normally she can download 2 GB/minute, but 40% of the way through the download, Windows forces a restart to install updates, which takes 20 minutes. Then Carla has to restart the download from the beginning. How long does it take to download the file?*
**Answer:** Carla can download 2 GB/minute. The file is 200 GB. So it will take 200/2 = 100 minutes to download the file. 40% of the way is 40% * 200 = 80 GB. So it takes 80/2 = 40 minutes to download 80 GB. Then it takes 20 more minutes to install updates. Then Carla has to restart the download from the beginning. So it takes 40 + 20 = 60 minutes to download 80 GB. The rest of the file is 200 - 80 = 120 GB. So it takes 120/2 = 60 minutes to download 120 GB. In total it takes 60 + 60 = 120 minutes. **The answer is 120 minutes**.

---

### B.1  EXAMPLES OF REFLECTION

## C  PROMPTS

---

**Example 1: Slow-thinking generation (GSM8k; Llama-3-8b) $\alpha$ is set to 0.12**

*Query*: *Carla is downloading a 200 GB file. Normally she can download 2 GB/minute, but 40% of the way through the download, Windows forces a restart to install updates, which takes 20 minutes. Then Carla has to restart the download from the beginning. How long does it take to download the file?*

**Answer:** Let's break down the problem step by step.

Initially, Carla can download the 200 GB file at a rate of 2 GB/minute. To find the time it would take her to download the entire file, we can set up a proportion:

Time to download entire file = Total size of file / Download rate

= 200 GB / (2 GB/minute)

= 100 minutes

Now, let's consider what happens when Windows forces a restart 40% of the way through the download. Since Carla has downloaded 40% of the file, she has already downloaded 0.4 * 200 GB = 80 GB of the file. The remaining amount to be downloaded is 200 GB - 80 GB = 120 GB.

The restart takes 20 minutes, during which time Carla does not download any additional data. Once the restart is complete, Carla has to restart the download from the beginning, effectively losing all the progress she made so far.

To find the time it takes Carla to download the file after the restart, we can set up another proportion:

Time to download file after restart = Size of file / Download rate = 200 GB / (2 GB/minute) = 100 minutes.

Since Carla has to restart the download from the beginning, she effectively has to download the entire file again. We can combine the time it takes her to download the entire file with the time it takes her to download the initial file:

Total time to download file = Initial time to download partial file + Time to restart + Time to download the file

= 40 minutes + 20 minutes + 100 minutes

= 160 minutes

Therefore, it takes Carla a total of **160 minutes**, or 2.67 hours, to download the 200 GB file

---

**Example 1: Fast-thinking generation (GSM8k; Llama-3-8b) $\alpha$ is set to -0.12**

*Query*: *Carla is downloading a 200 GB file. Normally she can download 2 GB/minute, but 40% of the way through the download, Windows forces a restart to install updates, which takes 20 minutes. Then Carla has to restart the download from the beginning. How long does it take to download the file?*

**Answer:** The total time of downloading the file is **120 minutes**.

---

**Example 2: Original generation (GSM8k; Llama-3-8b) $\alpha$ is set to 0**

*Query*: *Melanie is a door-to-door saleswoman. She sold a third of her vacuum cleaners at the green house, 2 more to the red house, and half of what was left at the orange house. If Melanie has 5 vacuum cleaners left, how many did she start with?*

**Answer:** Melanie has 5 vacuum cleaners left. She sold half of what was left at the orange house. So she sold 5 / 2 = 2.5 at the orange house. Then she sold 2 more at the red house. So she sold 2.5 + 2 = 4.5 at the orange and red houses. Then she sold 1/3 of her vacuum cleaners at the green house. So she sold 1/3 * x = 4.5 - 2.5 = 2 at the green house. So x = 6. The answer is 6.

---

**Example 2: Slow-thinking generation (GSM8k; Llama-3-8b)** $\alpha$ **is set to 0.11**

*Query*: *Melanie is a door-to-door saleswoman. She sold a third of her vacuum cleaners at the green house, 2 more to the red house, and half of what was left at the orange house. If Melanie has 5 vacuum cleaners left, how many did she start with?*
**Answer:** Let's break this problem down step by step!

Let's say Melanie starts with x vacuum cleaners initially.

At the green house, she sells a third of her vacuum cleaners, which is x/3. So, she has x - x/3 = 2x/3 vacuum cleaners left after selling at the green house.

Next, she sells 2 more vacuum cleaners at the red house, leaving her with 2x/3 - 2 vacuum cleaners.

Finally, she sells half of what's left at the orange house. Since she has 2x/3 - 2 vacuum cleaners left, half of this amount is (2x/3 - 2)/2 = x/3 - 1 vacuum cleaners.

Now, we're told that Melanie has 5 vacuum cleaners left after all these sales. We can set up an equation using the information above:

x/3 - 1 = 5

To solve for x, we can multiply both sides of the equation by 3 to get rid of the fraction:

x - 3 = 15. Then, add 3 to both sides to isolate x:

x = 18

So, Melanie started with 18 vacuum cleaners initially.

**Example 2: Fast-thinking generation (GSM8k; Llama-3-8b)** $\alpha$ **is set to -0.14**

*Query*: *Melanie is a door-to-door saleswoman. She sold a third of her vacuum cleaners at the green house, 2 more to the red house, and half of what was left at the orange house. If Melanie has 5 vacuum cleaners left, how many did she start with?*
**Answer:** The answer is 5.

**Reflection Example: Initial generation (GSM8k; Llama-3-8b)**

*Query*: *Candice put 80 post-it notes in her purse before she headed out to her job at the coffee shop. On her way, she stopped off at the store and purchased a package of Post-it notes; At work, she placed a single Post-it note on each of 220 different cups of coffee. If she had 23 post-it notes remaining overall, how many Post-it notes were in the package that she purchased?*
**Initial generation:** Candice started with 80 Post-it notes. After placing 220 on coffee cups, she had 80 - 220 = -140. So she must have had -140 in the purse, but she had 23 left. So she must have had -140 + 23 = -117 in the purse. She started with 80, so she must have had 80 - 117 = -37 Post-it notes in the purse. At work, she placed 220 on coffee cups. So she must have had 80 + 220 = 300 Post-it notes overall. She had 23 left, so she must have had 300 - 23 = 277 Post-it notes at work. She had 277 - 80 = 197 Post-it notes at work. 197 - 220 = -23. So she must have had -23 Post-it notes in the purse. She started with -117, so she must have had -117 + 23 = -94 Post-it notes in the purse. So she must have had 94 Post-it notes in the package that she purchased. **The answer is 94**.

**Reflection Example: too strong strength generation (GSM8k; Llama-3-8b) $\alpha$ is set to 1**

**Reflection:** Let's review the existing solution steps:
1. Candice started with 80 Post-it notes. 2. She placed a single Post-it note on each of 220 different cups of coffee, so she used 220 Post-it notes. 3. She had 23 Post-it notes remaining overall.
The issue with the existing solution steps is that it doesn't accurately calculate the number of Post-it notes Candice had before placing them on the coffee cups. The correct calculation should be:
80 (initial Post-it notes) - 220 (used for coffee cups) = -140
This means Candice had -140 Post-it notes before placing them on the coffee cups.
The existing solution steps then incorrectly calculate the number of Post-it notes in the package Candice purchased:
-140 (initial Post-it notes) + 23 (remaining Post-it notes) = -117
However, this is not the correct calculation. Instead, we need to find the number of Post-it notes Candice had overall and subtract the number of Post-it notes she had initially to get the number of Post-it notes in the package:
Total Post-it notes = 23 (remaining Post-it notes) + 80 (initial Post-it notes) + 220 (used for coffee cups) = 323
Number of Post-it notes in the package = Total Post-it notes - Initial Post-it notes = 323 - 80 = 243
**Therefore, Candice had 243 Post-it notes in the package she purchased**.

**Reflection Example: appropriate strength generation (GSM8k; Llama-3-8b) $\alpha$ is set to 0.5**

**Reflection:** I think I see some issues with the previous solution steps.
Let's re-evaluate the problem:
Candice starts with 80 Post-it notes.
At work, she places a single Post-it note on each of 220 coffee cups. That's a total of 220 Post-it notes used. She has 23 Post-it notes remaining overall. To find the total number of Post-it notes she had at the end, we add the remaining 23 to the 220 used at work: 23 + 220 = 243.
Now, let's find the number of Post-it notes in the package she purchased. She started with 80 and had 243 at the end. To find the number of Post-it notes in the package, we subtract the initial 80 from the total 243: 243 - 80 = 163.
So, Candice purchased a package of 163 Post-it notes.
**The previous solution steps had some errors, but we've corrected them to arrive at the correct answer: 163 Post-it notes in the package**.

**Reflection Example: too weak strength generation (GSM8k; Llama-3-8b) $\alpha$ is set to 0**

**Reflection:** Candice started with 80 post-it notes before she left for work. At work, she placed 220 on coffee cups. So at the end, she had 80 - 220 = -140 post-it notes. After she had 23, she had -140 + 23 = -117 post-it notes in her purse. At work, she had 80 + 220 = 300 post-it notes. She had 300 - 23 = 277. At work, she had 277 - 80 = 197 post-it notes. 197 - 220 = -23. She had -23 post-it notes in the purse. She started with -117, so she had -117 + 23 = -94 post-it notes in the purse.
**She had 94 post-it notes in the package that she purchased**.

**Prompt for answer formatting (GSM8k)**

Generate the final answer for the query {query} based on the reasoning process reasoning in the format: "Answer: ", followed by your numerical answer, which should be an integer without ',' or other symbol. Do not include any other text.

**Prompt for answer formatting (MATH500)**

Generate the final answer for the query {query} based on the reasoning process {reasoning} in this format: "Answer: \[ \boxed{{your answer here}} \]". The entire answer should be contained completely within the \boxed{{}} command. Do not include any other text.

**Prompt for answer formatting (GPQA)**

Generate the final answer for the query: {query} based on the reasons: {reasoning}. The final answer must be in this format: "Answer: A/B/C/D" (e.g. "Answer: A"). Do not include any other text.

