# OpenReview forum: "Fractional Reasoning via Latent Steering Vectors Improves Inference Time Compute"
_ICLR.cc/2026/Conference — Submitted to ICLR 2026_

### Official Review · Reviewer_yNjN · 2025-10-29

**Soundness:** 3
**Presentation:** 3
**Contribution:** 3
**Rating:** 4
**Confidence:** 3

**Summary:**

The paper proposes a training-free framework for controlling the reasoning intensity of models at inference time. The framework involves (1) computing a latent steering vector based on multiple pairs of unlabelled data with and without CoT, (2) identifying the first principle component of these difference vectors, (3) add this vector to the original model's latent state scaled by a tunable factor $\alpha$. Given this tunable $\alpha$, an ensemble of responses with different $\alpha$ can then be produced and aggregated over to obtain better reasoning performance.

**Strengths:**

- The proposed concept of having a continuous tunable parameter to control reasoning intensity via latent space arithmetic is an interesting and innovative concept.
- The method is training-free and has been demonstrated to perform well compared to basic baselines, across various datasets and models.
-  There is analysis of the impact of $\alpha$ to model output, quantitatively through its correlation with generation length (Fig. 4) and some qualitative examples.
- The paper applied the approach to two different test-time strategies -- CoT and self-reflection strategies.

**Weaknesses:**

- The main motivation of the paper does not match the proposed intervention and experimental design. The authors proposed the ability to have tunable reasoning intensity as different questions may require different levels of reasoning intensity, but subsequently in the paper Sec.3 mentioned that it is inherently difficult to identify ideal prompt strength per question. The rest of the paper then discussed ensemble-based methods to improve reasoning with a uniform distribution of $\alpha$ to form the distribution, which is a different problem setting. It would be useful if the paper could be either repositioned, or include more experimental results specifically demonstrating how $\alpha$ is directly related to reasoning intensity as claimed.

- The paper compares against fairly basic LLM ensemble methods for improved reasoning (self-consistency with majority vote, best of N). The authors should consider discussing in related work and/or comparing with methods [1-3] that propose LLM prompt ensembles with improved reasoning performance since those are most directly comparable to the proposed Fractional Reasoning framework (it also effectively creates an ensemble with different prompts). It is unclear if the reported performance gains may stem mainly from more diverse 'prompting' of the LLM ensemble.

- The method requires first computing the latent steering vector based on the principle components of 'queries', though it is unclear from the paper how many or what type of queries would be required (held-out set of the same question distribution?). Providing greater clarity on this would help address concerns regarding generalizability of the method or the cost of applying such methods if the latent steering vectors are not general.
   - Explicitly showing whether the latent steering vectors are generalizable across tasks and datasets would be useful. Related to this is providing a more thorough description (including what queries and how many are needed, from where) of the specific pipeline used for computing the steering vectors.

- The computational costs of the method seem large, especially compared to the basic baselines proposed to be used. The authors should consider reporting a more thorough analysis of the computational cost overheads compared to other methods, especially since the original motivation of the paper relates to efficiency. This includes the cost of generating the latent steering vectors along with vector addition and scaling at every token and layer.

- The change in methodology for Reflection seems ad-hoc, weakening the paper's unified framework. It is not clear why for Reflection (compared to CoT), the contrastive-pair approach should be changed (similarly why the single prompt latent approach should not be applied for the CoT setting). The authors should provide further elaboration and analysis on why there is this lack of consistency.

- The $\alpha$ hyperparameter seems relatively important. While there has been some ablation done, it would be helpful if the authors could further elaborate on whether this needs to be tuned for different model and task, and whether there is some intuition behind the selection of the range if it is important.



[1] Hu et al, Dipper: Diversity in Prompts for Producing Large Language Model Ensembles in Reasoning Tasks

[2] Chen et al, AgentVerse: Facilitating Multi-Agent Collaboration and Exploring Emergent Behaviors.

[3] Du et al, Improving factuality and Reasoning in Language Models through Multiagent Debate

**Questions:**

Please see concerns and questions in the Weakness section.

---

> ### Author Response · Authors · 2025-11-24
>
> We thank the reviewer for their detailed review. We are glad that the reviewer highlights the novelty, the strong empirical performance and the broad applicability of our work and commends the analysis section. We address each concern below.
>
> ---
>
> ## **Relation between Motivation and Method (W1)**
>
> The reviewer notes there seems a disconnection between the motivation and the method. We thank the reviewer for highlighting this, and we first elaborate on the relation between them, then revise the paper to make the connection clearer as the reviewer suggests.
>
> The reviewer notes that the motivating example suggests "different questions may require different levels of reasoning intensity". In fact, our method is the *solution* to the motivation, not a departure from it.
>
> While directly finding an oracle reasoning intensity ($\alpha$) for each question is a straightforward way to address the problem, as we state in the paper, finding the ideal $\alpha$ *a priori* is "inherently difficult".
>
> Instead, we proposed an alternative method that still address the problem by providing different levels of reasoning intensity:
>
> 1. Generate candidates that span different levels of reasoning intensity.
> 2. Use an established test-time compute method (like Best-of-N or Majority Vote) to ***select*** the best answer from this diverse pool.
>
> We'd like to highlight that the difference between our method and a standard ensemble (e.g., via temperature sampling as the baselines in our experiments) is that standard ensemble creates stochastic variations of the ***same* reasoning intensity**, while our method creates a *structured diversity across **different levels reasoning intensity***. Our method works better because by exploring a range of different levels reasoning intensity, we increase the overall probability of generating the correct answer within our candidate pool. The verifier (PRM or majority vote) then effectively identifies the best $\alpha$ *post-hoc*. This *directly* addresses the motivation: our method leverages different levels of reasoning intensity to achieve better performance.
>
> Our method is not just a simple ensemble. Instead of developing a thoughtful algorithm to determine the reasoning intensity ($\alpha$) *a priori*, our method use well-known methods like Best-of-N or Majority Vote to select the reasoning intensity ($\alpha$) *post-hoc*.
>
> Our method provides different levels of reasoning intensity and use established test-time method (like Best-of-N or Majority Vote) to ***select*** **the best reasoning intensity**, which **aligns with the motivating example** which suggests "different questions may require different levels of reasoning intensity".
>
> We've revised the paper to make this connection between our motivation and our method explicit. We will clarify that FR is a *candidate generation* strategy that creates a more diverse and high-quality solution space for *any* test-time selection method.
>
> ---
>
> ## **Comparison to other LLM Ensemble Methods (W2)**
>
> The reviewer suggests we should compare to other ensemble methods and provided references[1-3]. We thank the reviewer for these references and have cited all of them in Section 5.4. Inspired by [1] (Dipper), we added an experiment with prompting ensemble baselines. Specifically, we use the following prompts:
>
> - "Solve the problem with step-by-step detailed reasoning"
> - "Solve the problem with direct answering"
> - "Solve the problem with step-by-step detailed reasoning, think harder"
> - "Solve the problem with step-by-step reasoning, think less"
> - "Solve the problem"
>
> **GSM8K, Llama-3 Results:**
>
> | Method                               | Accuracy |
> | :----------------------------------- | :------- |
> | Majority Vote                        | 86.9     |
> | Majority Vote + Diverse Prompts      | 85.2     |
> | Majority Vote + Fractional Reasoning | **89.5** |
>
> Textual prompting diversity **underperforms** our fractional reasoning. This highlights the **limitation of prompt engineering** for controlling reasoning and strengthens the advantages of our framework.
>
> In addition, [2] (AgentVerse) and [3] (Multiagent Debate) describe complex multi-agent collaboration frameworks, which are a different paradigm from our work's focus. We've cited both of them and provided a discussion paragraph on potential future direction of applying our FR framework to multi-agent setting.
>
> [1] Hu et al, Dipper: Diversity in Prompts for Producing Large Language Model Ensembles in Reasoning Tasks
>
> [2] Chen et al, AgentVerse: Facilitating Multi-Agent Collaboration and Exploring Emergent Behaviors.
>
> [3] Du et al, Improving factuality and Reasoning in Language Models through Multiagent Debate

---

> ### Author Response · Authors · 2025-11-24
>
> ## **Clarification on Steering Vector Computation (W3)**
>
> The reviewer asks questions regarding details of steering vector computation. First, they ask how many or what type of queries would be required. We set the number of queries needed to 256 in our experiments. In addition, we've conducted an ablation study that examines how the number of queries influences the final performance, with results shown in Table 4 of Section A.3. The performance is extremely robust across different number of queries needed. For the type of the queries, we don't require specific types and just use the original queries from the dataset.
>
> Second, the reviewer asks about generalizability and cost for applying the latent steering vectors. We'd like to highlight the facts below:
>
> - Computing the steering vector is a **single forward pass** — no backpropagation, fine-tuning, or reward model training.
> - In practice, it adds **<1% inference-time overhead** on GSM8K, MATH-500 and GPQA and can be fully parallelized.
> - The steering vector for each dataset and model could be **precomputed once and saved** for all future use cases.
> - This is orders of magnitude lighter than training-intensive baselines such as RLHF or PEFT, yet delivers **consistent performance gains** across all tested models and benchmarks.
>
> In short, the cost is negligible relative to the accuracy improvements we achieve. The method applies to any model and any dataset.
>
> ---
>
> ## **Clarification on Computational Cost (W4)**
>
> The reviewer is concerned about computational overhead, including latent steering vector generation and the per-token operations.
>
> - Regarding overhead of latent steering vector generation, as is addressed in W3 above, the computational overhead is negligible. In practice, it adds **<1% inference-time overhead**.
> - Regarding overhead of per-token operations, our method consists of only one vector addition ($h_t := h_t + \alpha h_{steer}$) and a norm-rescaling operation (devide the vector by a constant). These operations are computationally **trivial** compared to the massive matrix multiplications in the Transformer's attention and MLP layers. The wall-clock time for a single forward pass with FR is **almost identical** to a standard forward pass. In practice, the time for forward pass of our method and baseline is 1.012:1.
>
> In short, the computational costs of our method and baselines are nearly the same.

---

> ### Author Response · Authors · 2025-11-24
>
> ## **Rationale for Different Instantiations (W5)**
>
> The reviewer suggests further elaboration for why two different instantiation methods are used in our framework. We'd like to clarify that this is a **deliberate, principled design choice** instead of an ad-hoc one.
>
> We'd like to note that this choice stems from the fundamental difference between the two tasks:
>
> 1. **CoT is a General Behavior:** We want to apply a general, **global** behavioral shift (i.e., "think step-by-step" or "direct answering"). This desired behavior is *not* specific to any single query.
>
>    Therefore, we compute a single, static vector by averaging over many contrastive examples.
>
> 2. **Reflection is an Instance-Specific and Context-Specific Intervention:** We want the model to critique a *specific* answer to a *specific* query. The steering *must* be context-specific (it must take the original answer into account). The input context is therefore much longer and more semantically complex (reflection prompt + original query + initial answer).
>
>    Therefore, the steering signal must be **dynamically tailored to the specific content** being reviewed. For this reason, our method for reflection directly computes the latent shift induced by the reflection prompt *for that specific instance*.
>
> In short, the **CoT vector is a static, global prior** on reasoning style, while the **reflection vector is a dynamic, local intervention** based on the immediate context (initial answer).
>
> The choice of instantiation demonstrates the flexibility and generalizability of our framework. Fractional Reasoning provides a **general framework for controlling model behavior via latent steering**; it **does not rely on a single, fixed instantiation** to be effective. Researchers can choose the most appropriate method for constructing the steering vector (e.g., global vs. instance-specific) depending on the problem they are tackling.
>
> As this is a critical point of confusion, we will add a new paragraph to the paper to explicitly discuss this rationale and explain why this flexibility is a *strength* of our framework.
>
> ---
>
> ## **Elaboration on Hyperparameter $\alpha$ (W6)**
>
> The reviewer asks for more elaboration on the $\alpha$ range, how it's selected, and if it needs tuning.
>
> - **Tuning:** We use the **same range** $[-0.15, 0.15]$ for all models and tasks in our experiments. It does not require any meticulous per-task and/or per-model tuning.
> - **Robustness & Intuition:** Our ablation in **Figure 6** in Section A.3 directly investigates this. The first intuition is that extremely large $\alpha$ would not yeild desired generation as it steer the latent states too much, and in practice we find that setting $\alpha$ to $[-0.2,0.2]$ is able to avoid such phenomenon across different models and tasks. Then, within $[-0.2,0.2]$, the key intuition is that the range must be wide enough to introduce *sufficient diversity* (to generate both concise and detailed responses). Once that diversity is achieved,  performance remains roubust and stable, indicating that our method is robust to the choice of range once sufficient diversity in reasoning strength is introduced. Such rule applies to different models and tasks. Therefore, no specific tunning is needed in our experiments.
>
> ---
>
> In summary, we've clarified the connection between our motivation and method, added experiments to include prompt ensemble baselines, clarified steering vector computation detail and minimal computational overhead of our method, provided rationale for different instantiation choices and hyperparameter selection. We hope these clarifications resolve the reviewer's concerns and welcome any further discussion.

---

### Official Review · Reviewer_wwH8 · 2025-10-31

**Soundness:** 3
**Presentation:** 3
**Contribution:** 3
**Rating:** 6
**Confidence:** 3

**Summary:**

This paper proposes Fractional Reasoning (FR), a training-free, model-agnostic framework that provides continuous, inference-time control over an LLM’s “reasoning intensity” by steering internal latent states. The core idea is to extract a steering vector that captures the latent shift induced by reasoning-promoting prompts (e.g., CoT or reflection) and then reapply this vector with a tunable scaling factor α during generation. This enables adaptive control of depth: from terse direct answers (low α) to richer multi-step reasoning and stronger reflection (higher α), without modifying input text or fine-tuning the model.

Methodologically, the paper formalizes prompts as latent state shifts in attention and constructs a steering vector via contrastive pairs (positive: “step-by-step reasoning” vs negative: “direct answering”), selecting the first principal direction of the difference. The steered latent states are norm-rescaled for stability. The framework supports two test-time scaling modes: breadth-based (e.g., Best-of-N, Majority Vote) by introducing structured diversity across α values, and depth-based (reflection) by controlling reflection strength. Experiments on GSM8K, MATH500, and GPQA with LLaMA-3-8B-Instruct and Qwen-2.5-7B-Instruct show consistent gains over standard prompting and over baseline test-time compute; results generalize to a reasoning-tuned model (DeepSeek-R1-Distill-Qwen-7B). Additional analyses show monotonic control over verbosity with α, favorable scaling with the number of generations, reflection improvements, robustness to the number of contrastive pairs, and saturation in α-range ablations. A sentence-level variant demonstrates dynamic adjustment of α based on feedback (e.g., PRM/consistency), correcting errors that instance-level control misses.

**Strengths:**

The paper introduces a simple, principled, and interpretable mechanism for inference-time control of reasoning via latent steering vectors, avoiding input rewrites or fine-tuning. It clearly connects instructional prompts to directional latent shifts and operationalizes this with a normalized additive intervention that is easy to implement and stable. The framework is broadly applicable: it enhances both breadth-based strategies (improving the quality/diversity of candidate generations for Majority Vote and Best-of-N) and depth-based strategies (controlling reflection strength, mitigating both under- and over-reflection). Empirically, results are consistent across multiple datasets and two strong open-source models, and even on a reasoning-specialized model. The analysis section is thoughtful: it demonstrates behavioral control (generation length vs α), scalable benefits with more generations, ablations on α ranges and number of contrastive pairs, and a compelling step-level reflection case. Overall, the contribution is timely and practically significant for the growing area of inference-time scaling.

**Weaknesses:**

While the contrastive construction of the steering vector is justified and practical, the paper would benefit from deeper analysis of where and how to intervene (per-layer/per-position granularity, keys/values/queries, attention vs MLP streams). The current choice (last-token representations concatenated across layers) is one of many possible designs; an ablation across layers or modules could strengthen claims of generality and inform best practices. The Best-of-N results hinge on an external PRM; more scrutiny of PRM sensitivity (e.g., swapping PRMs, PRM calibration vs α-induced diversity) would isolate the contribution of FR under different reward landscapes. Robustness and safety aspects are lightly treated: although inputs are public, the method changes internal activations; a brief discussion on failure modes (instability, hallucination shifts, adversarial α ranges) and safeguards would be useful. Finally, the dynamic (sentence-level) α control is presented as a case study; quantitative results comparing instance-level vs sentence-level control across benchmarks would substantiate its impact.

**Questions:**

How sensitive are the gains to the locus of intervention? For example: steering only specific layers (early/mid/late), only attention vs only MLP, or steering keys/values/queries separately. A systematic ablation could reveal more efficient and robust placements and perhaps reduce compute overhead. Can the authors quantify a compute-efficiency curve: accuracy vs total tokens/forward passes under fixed budgets, comparing FR-enhanced Majority Vote/Best-of-N against stronger baselines (e.g., self-consistency with calibrated temperatures, diversified prompts, or MC-SC)? This would clarify the “accuracy-per-sample” and “accuracy-per-token” advantages. In the reflection setting, can the sentence-level dynamic α be evaluated at scale with a simple feedback signal (e.g., PRM score thresholds) to provide a quantitative gain over instance-level reflection across GSM8K/MATH500/GPQA? Even a modest study (e.g., 1–2k instances) would help establish its general benefit. More broadly, can FR steer away from known pitfalls (e.g., spurious long CoT that sounds plausible but is wrong) by combining α control with lightweight detectors?

---

> ### Author Response · Authors · 2025-11-24
>
> We thank the reviewer for highlighting our framework as "principled and interpretable mechanism" and "practically significant contribution" that is "broadly applicable" to various tasks, and for commending our consistently strong empirical results and thoughtful analysis section. We address each concern below.
>
> ---
>
> ## **More Analysis on the Locus of Intervention**
>
> The reviewer suggests providing more analysis on the design choice of locus of intervention. We agree that a deep ablation on the locus of intervention is an important research question. However, we want to clarify that the main focus of our work is to propose a **general framework**, Fractional Reasoning, to quantitatively steer reasoning behavior rather than to find the single optimal instantiation. Our chosen instantiation was inspired by prior work and our results demonstrate the effectiveness at achieving this control, as the reviewer mentioned in the strength section.
>
> We believe our framework remains flexible, and researchers can choose their own instantiations as the reviewer pointed out to best suit their specific tasks. We appreciate the reviewer pointing this out, as it highlights the generality of our approach.
>
> We have added a paragraph in Section 5.4 to acknowledge this. We explicitly state that our method is one possible instantiation, that the framework is flexible, and that a deep dive into the optimal locus of intervention is a promising direction for future work.
>
> ---
>
> ## **More Analysis on PRM for Best-of-N**
>
> The reviewer suggests more analysis on the PRM used for Best-of-N setting. We agree that this is an important point. We chose a strong publicly available PRM to demonstrate that our framework works well within the widely-used Best-of-N setting. A deep analysis of PRM sensitivity is a valuable but distinct research direction. Our framework is flexible and researchers are free to pair FR with any PRM of their choice.
>
> Crucially, to **isolate the effect of our work** without *any* external dependencies, we also provided extensive experiments using **Majority Vote**. This verifier-free setting demonstrates that FR consistently improves performance on its own, by fundamentally shifting the distribution of generated outputs to be of higher quality and diversity.
>
> We have added a paragraph in Section 5.4 to explicitly discuss this.
>
> ------
>
> ## **Discussion on Safety**
>
> The reviewer suggests adding discussion on safety aspects. We sincerely thank the reviewer for raising this critical point. This is an important consideration and we have updated the "Ethics Statement" section of the paper, and we discuss the potential risks of malicious use of the framework. We believe addressing broader societal impacts provides a more complete and responsible discussion, and we are grateful to the reviewer for bringing up this addition.
>
> ---
>
> ## **Sentence-level Control**
>
> The revirewer suggests additional results with sentence-level control. We want to clarify that this sentence-level control was presented not as a main quantitative result, but as a **proof-of-concept** to demonstrate the *versatility and future potential* of the FR framework. A full-scale quantitative study of this dynamic control would be a significant follow-up paper.
>
> ---
>
> ## **Compute-efficiency Curve**
>
> The reviewer's request for a compute-efficiency curve is an excellent one. We believe **Figure 5** directly addresses this. This figure plots **accuracy as a function of the number of generations**.
>
> Since the number of generations is the primary driver of compute cost (tokens/forward passes) in test-time scaling, this figure serves as a direct compute-efficiency curve. It clearly shows that for a *given compute budget*, our Fractional Reasoning consistently achieve higher accuracy than their respective baselines. This validates that FR provides a more efficient and robust use of inference-time compute.
>
> ---
>
> ## **Avoid Known Pitfalls of Spurious Long CoT**
>
> The reviewer asks if our FR can avoid known pitfalls of spurious long CoT but is wrong. This is a fantastic question, and the answer is **yes**.
>
> This is one of the most promising applications of our framework. By setting $\alpha$ to **negative values**, we can actively steer the model away from "over-thinking" and reduce the generation of spurious, long reasoning. We find this is particularly effective for complex, reasoning-tuned models (like the one used in Section 5.2), which are more prone to this failure mode.
>
> ------
>
> We thank the reviewer again for their feedback. We hope these clarifications address the reviewer’s concerns and are more than happy to engage in further discussion.

---

### Official Review · Reviewer_v6Yx · 2025-11-01

**Soundness:** 2
**Presentation:** 3
**Contribution:** 2
**Rating:** 2
**Confidence:** 4

**Summary:**

The paper introduces a  training-free framework that effectively improves test-time compute by enabling fine-grained, continuous control over a model's reasoning intensity via latent steering vectors.

**Strengths:**

1. The paper is well-written,  and easy to follow .
2. It presents a clear motivation and supports its claims with relatively thorough experimental validation.

**Weaknesses:**

1. The core technique is a direct application of existing work (e.g., Representation Engineering, Activation Addition). The paper applies a known method to a new axis ("reasoning vs. direct answer") and lacks a fundamental methodological contribution. It is more of an application case study than novel research. Moreover, several existing papers (e.g. [1]) present ideas that are highly similar to those in this work.
2. The paper wrongly assumes "reasoning" is a single, linear dimension controllable by a scalar α. This is a flawed model of a complex process. The evidence (e.g., Figure 4 showing correlation with length) strongly suggests the method is controlling verbosity, not true logical depth. Performance gains may just be an artifact of longer, more verbose outputs.
3. The fairness of the comparison between the FR method (using N different α  values) and the baseline (sampling N times from 1 fixed prompt) is worth further consideration. By design, the FR method introduces a form of "strategy ensemble" (mixing different reasoning intensities), whereas the baseline lacks this strategy-level diversity. As a result, it is difficult to disentangle from the current results how much of the performance lift comes from the merits of "fractional control" itself versus this built-in ensemble effect.
4. The paper's claims regarding "adaptivity" seem somewhat overstated. In the current implementation, the model itself does not appear to dynamically match its reasoning intensity to a specific problem's difficulty during generation. Rather, it seems to perform a non-adaptive search across N different reasoning styles, with the "intelligence" or "adaptivity" being deferred to the expensive post-processing selection step (e.g., Best-of-N or voting).
5. The score for the DeepSeek model in Table 3 appear to contain a significant and counter-intuitive anomaly. The paper reports a score of 78.6% on GSM8K (relatively simple) but 92.4% on the much harder MATH500. This result is highly unusual, as the difficulty of MATH500 far exceeds that of GSM8K (a fact supported by the paper's other data and the wider literature).

[1] Controlling thinking speed in reasoning models

**Questions:**

Please see the weaknesses.

---

> ### Author Response · Authors · 2025-11-24
>
> We thank the reviewer for their feedback. We are glad they found the paper "well-written" with "clear motivation" and "thorough experimental validation". We address each concern below.
>
> ---
>
> ## **Contribution (W1)**
>
> The reviewer states potential similarity with existing work. We respectfully disagree.
>
> We agree that we build upon the *foundational mechanism* of latent steering, as pioneered by works like Representation Engineering. This is a powerful tool, just as "convolution" or "attention" are powerful tools. However, our novelty lies not in the base mechanism (vector addition). Our novel contributions are:
>
> - **A New, Complex Control Axis:** Prior work focused on steering simpler, low-level attributes (e.g., sentiment, topic). Our paper is the first to demonstrate that a complex, emergent, high-level cognitive property like "reasoning intensity" and "reflection strength" is *also* linearly represented and controllable via a single steering vector. This is a non-trivial scientific finding about the model's internal state, not just an "application."
>
> - **A General Framework:** We show this is a *general framework* by applying it to *two distinct* test-time compute paradigms: **breadth-based CoT** and **depth-based Reflection**. This unified application is novel and demonstrates the generality of our FR framework.
>
> We also thank the reviewer for bringing up the excellent work [1], which we've cited and discussed the distinction:
>
> - **Different Core Methodology:** Our central methodological contribution is *not* just the steering vector. It is the concept of **Fractional Reasoning framework**: using a diverse set of reasoning intensity to create a candidate pool that increases the overall probability of generating the correct answer. Steering vector is just one example instantiation of our FR framework. We've also showed other instantiation such as latent states intropolation which is not used in [1].
>
> - **Broader Framework:** We present a *general framework* for improving test-time compute. We demonstrate this by successfully applying it to two *distinct* test-time strategies: **breadth-based CoT**  and **depth-based Reflection**. This unified application to both reasoning and reflection is a novel contribution.
>
> In short: [1] is an excellent work that explores how to build a steering vector for controlling the thinking speed in reasoning models, while our paper provides a novel framework for how to define a unified framework to **systematically improve test-time scaling**. This is a new, orthogonal contribution, not a mere application.
>
> [1] Lin, Zhengkai, et al. "Controlling Thinking Speed in Reasoning Models." *The Thirty-ninth Annual Conference on Neural Information Processing Systems*.

---

> ### Author Response · Authors · 2025-11-24
>
> ## **Assumption and Reason for Performance Gain (W2)**
>
> The reviewer asserts that we assume reasoning is a single, linear dimension, and that Figure 4 suggests performance gains may just be an "artifact of longer, more verbose outputs". We respectfully disagree with this characterization on both fronts.
>
> - **On Linearity:**  Our method does not assume "reasoning" is a *perfectly* linear dimension. Inspired by previous works (e.g., [2,3]) that provide theoretical justification that latent representations in LLMs exhibit linear structure, our method shows that latent features are sufficiently coherent for simple vector arithmetic to steer high-level behavior. **Our strong empirical results showing consistently improved performance** across model architectures (Qwen, Llama), types (general and reasoning), and benchmarks (GSM8K, MATH500, GPQA) are the ultimate validation that this linear steering approach is practically effective and robust.
>
> - **On Verbosity vs. Reasoning:** We provide the following evidence to resolve the reviewer's primary concern that Figure 4 suggests our gains are an artifact of "longer, more verbose outputs":
>
>   - **Negative $\alpha$ (Shorter Answers) Improves Performance:** Our method is not about *only* making outputs longer. A key part of our ensemble is using **negative $\alpha$ values** to generate *shorter, more concise* answers. This is especially true for strong reasoning models (Table 3), which are already verbose and can benefit from being steered *away* from "over-thinking." This refutes the "longer and verbose brings better performance" hypothesis.
>   - **Figure 4 is not a measure of performance**. It is a behavioral analysis intended to show that our control mechanism is *interpretable*: $\alpha$ has the expected monotonic correlation with reasoning depth, as deeper reasoning naturally requires more steps and words. Generation length is a good metric to visualize the reasoning depth.
>
>   In short, the improvement does not stem from longer outputs as is evidenced by the fact that negative $\alpha$ leads to shorter answer and better performance in our method.
>
> [2] Jiang, Yibo, et al. "On the Origins of Linear Representations in Large Language Models." *International Conference on Machine Learning*. PMLR, 2024.
>
> [3] Park, Kiho, Yo Joong Choe, and Victor Veitch. "The Linear Representation Hypothesis and the Geometry of Large Language Models." *Forty-first International Conference on Machine Learning*.
>
> ---
>
> ## **Fairness of comparison (W3)**
>
> The reviewer questions the fairness of comparison because our FR method uses a mixing different reasoning intensities while the baseline does not. We respectfully disagree with the premise of this concern.
>
> The core of "fractional control" lies in the fact that by offering different levels of reasoning intensity, we increase the overall probability of generating the correct answer within our candidate pool. Therefore, the baseline we're comparing with is under the setting where no varing intensity is provided. Similar type of comparisons is well-accepted by the ML community and is commonly used.
>
> In addition, we've conducted an additional experiment with prompting ensemble baselines to address the reviewer's concern. Specifically, we use the following prompts:
>
> - "Solve the problem with step-by-step detailed reasoning"
> - "Solve the problem with direct answering"
> - "Solve the problem with step-by-step detailed reasoning, think harder"
> - "Solve the problem with step-by-step reasoning, think less"
> - "Solve the problem"
>
> **GSM8K, Llama-3 Results:**
>
> | Method                               | Accuracy |
> | :----------------------------------- | :------- |
> | Majority Vote                        | 86.9     |
> | Majority Vote + Diverse Prompts      | 85.2     |
> | Majority Vote + Fractional Reasoning | **89.5** |
>
> Textual prompting diversity **underperforms** our fractional reasoning. This highlights the **limitation of prompt engineering** for controlling reasoning and strengthens the advantages of our framework.

---

> ### Author Response · Authors · 2025-11-24
>
> ## **Adaptivity of Our Method (W4)**
>
> The reviewer claims our "adaptivity" is overstated, as the intelligence is in the expensive post-processing and not in the generation. Our response is twofold.
>
> 1. **On "Expensive Post-Processing":** We respectfully challenge the premise that our post-processing selection is "expensive" in this context.
>    - **Relative to an *a priori* method:** As we state in the paper, finding the *ideal* $\alpha$ *a priori* for a specific problem is "inherently difficult". Such an *a priori* algorithm would likely require a complex, auxiliary policy model, which could be *more* computationally expensive than the simple post-processing methods (like majority vote) we use.
>    - **Relative to training:** Our work is explicitly situated within the **test-time scaling** paradigm. The fundamental goal of this paradigm is to leverage additional inference-time compute to achieve better performance *without* the vastly more expensive alternative of retraining or fine-tuning the model. Therefore, the cost of post-selection is the *intended trade-off* of this entire research area and is not "expensive" when compared to the training-based alternative.
> 2. **On "Dynamic Adaptivity":** In addition to the "post-hoc adaptivity" (i.e., selection after generation), our framework *also* explicitly enables the dynamic, generative adaptivity the reviewer is asking for. We demonstrate this as a proof-of-concept in **Section 4 (Figure 3)**. This "Sentence-level control" example shows *exactly* this: a model that dynamically adjusts its reflection strength $\alpha$ *during* the generation process based on feedback signals from previous steps.
>
> ---
>
> ## **Scores in Table 3 (W5)**
>
> The reviewer is concerned about the scores in Table 3. We've provided full source code at the time of submission for reproducibility.
>
> ---
>
> In summary, we've highlighted our contribution beyond previous work, elaborated the reason for our performance gain is not from longer generation, showed the fairness of our comparision with baseline with additional prompt ensemble results, and explained adaptivity and scores in Table 3. We hope these clarifications address the reviewer’s concerns and are more than happy to engage in further discussion.

---

### Official Review · Reviewer_7WN4 · 2025-11-03

**Soundness:** 3
**Presentation:** 3
**Contribution:** 3
**Rating:** 6
**Confidence:** 3

**Summary:**

This paper proposes a framework called Fractional Reasoning, which enables adaptive control of a model’s reasoning depth and reflection strength during inference to improve the test-time computational efficiency of large language models (LLMs). By introducing a latent steering vector and a scaling factor, the method allows the model to dynamically adjust its reasoning behavior, striking a balance between concise direct answers and complex multi-step reasoning. Experimental results show that this approach significantly improves performance across multiple benchmarks, enhancing both the breadth and depth of reasoning while increasing sample efficiency. The framework is model-agnostic and can flexibly adapt reasoning strategies according to different task requirements and computational budgets.

**Strengths:**

1、The proposed method does not rely on model fine-tuning or modifying input texts, allowing it to be directly applied to existing LLMs. It supports a variety of LLMs, including both general instruction-tuned models and reasoning-specialized models, demonstrating strong generalizability.
2、The authors provide complete code and supplementary materials, ensuring high transparency and strong reproducibility, which enhances the credibility of the proposed framework.
3、The work is intuitively easy to understand and clearly explains how adjusting reasoning depth and reflection strength can control model behavior. By introducing latent steering vectors and tunable factors, the framework's design makes the reasoning process more controllable and easier to interpret.

**Weaknesses:**

1、Lack of explanation for the design of different latent steering approaches: The paper employs two different latent steering vectors for the reasoning (Chain-of-Thought) and reflection processes, but it does not clearly explain why these distinct designs are used. The absence of such an explanation weakens the interpretability of the framework and makes it difficult for readers to understand the specific impact of these design choices on model behavior.

**Questions:**

1、There are some minor errors in the writing process. In Section 5.2, the reference “As shown in Table ??” is incorrect; please fix this table citation.
2、Since two different latent steering vectors are used in the reasoning and reflection processes, it would be helpful to further explain in the paper why this design choice was made, so that readers can better understand it.

---

> ### Author Response · Authors · 2025-11-24
>
> We sincerely thank the reviewer for their positive feedback and insightful review. We are glad the reviewer recogonize our Fractional Reasoning framework as a "strong generalizable," "easy to interpret," and "controllable" approach, and we appreciate that they highlighted the paper's strong reproducibility. We address each concern below.
>
> ---
>
> ## **Rationale for Different Instantiations (W1, Q2)**
>
> The reviewer suggests additional explanation for why two different instantiation methods are used in our framework.
>
> We'd like to note that this choice stems from the fundamental difference between the two tasks:
>
> 1. **CoT is a General Behavior**: The goal of the CoT steering vector is to induce a **general, task-level behavior** (i.e., "think step-by-step" or "direct answering"). This desired behavior is *not* specific to any single query but is a global shift in the model's reasoning style.
>
>    Therefore, our method computes the steering vector using contrastive pairs, isolating a robust, **general-purpose "reasoning" vector** that can be applied to any new query to modulate the reasoning depth.
>
> 2. **Reflection is an Instance-Specific and Context-Specific Intervention:** In contrast, reflection is **highly instance-specific and context-specific**. The goal is to make the model critique and revise a *specific* initial generation for a specific query, meaning the intervention must depend on the initial answer. The input context is therefore much longer and more semantically complex (reflection prompt + original query + initial answer).
>
>    Therefore, the steering signal must be **dynamically tailored to the specific content** being reviewed. For this reason, our method for reflection directly computes the latent shift induced by the reflection prompt *for that specific instance*.
>
> In short, the **CoT vector is a static, global prior** on reasoning style, while the **reflection vector is a dynamic, local intervention** based on the immediate context (initial answer).
>
> The choice of instantiation demonstrates the flexibility and generalizability of our framework. Fractional Reasoning provides a **general framework for controlling model behavior via latent steering**; it **does not rely on a single, fixed instantiation** to be effective. Researchers can choose the most appropriate method for constructing the steering vector (e.g., global vs. instance-specific) depending on the problem they are tackling.
>
> We've updated the paper to explicitly discuss this rationale. We hope this will clarify the motivation and highlight the framework's flexibility, strengthening its interpretability as the reviewer suggested.
>
> ------
>
> ### **Minor Typo (Q1)**
>
> We thank the reviewer for catching this typo and have fixed it.
>
> ------
>
> We thank the reviewer again for their constructive and supportive review. We hope the clarification of the steering approaches rationale resolve the reviewer's concern and welcome further discussion.

---

### Meta-Review · Area_Chair_CKD4 · 2026-01-06

**Summary:**

This paper was reviewed by four experts in the field, with initial ratings 6/2/6/4, respectively. Reviewers raised concerns regarding the design rationale and the novelty, computational cost, clarity, adaptivity, and sensitivity of the proposed method. Some concerns have been addressed in the paper, but AC has checked the discussion and found some remain unaddressed. Such as the sensitivity (quantitative results would be better), adaptivity, and the computational cost (additional token length yielded by the method).

The AC notes that the authors used different values of max_new_tokens across models. For instance, 2048 for the base model and 8192 for the reasoning model, which inevitably leads to increased token cost. This overhead should be explicitly accounted for when evaluating efficiency. Besides, this hyperparameter may not suit more diverse scenarios with stronger reasoning models on challenging benchmarks.

Moreover, the set of evaluated models is limited. The study includes only Llama-3-8B-Instruct and Qwen-2.5-7B-Instruct as general models, and DeepSeek-R1-Distill-Qwen-7B as the reasoning model. The AC recommends including more capable and up-to-date models, such as those (8-32B) from the QwQ or Qwen-3 series, which demonstrate significantly stronger reasoning abilities.

Finally, the scope of the evaluation datasets is not convincing. In mathematical reasoning, standard benchmarks such as AIME 2024/2025 and Olympiad are commonly used to probe deep reasoning capabilities, and the GSM8K and MATH500 used may not sufficiently challenge strong reasoning models in this regard. Additionally, coding performance should be assessed using benchmarks like LiveCodeBench to verify the generalization capability of the proposed method.

Given ICLR’s competitive acceptance rate this year, the AC regrets to recommend rejection.

**Reviewer Concerns:**

- Rationale for design choices: Multiple reviewers questioned the use of different instantiations for CoT and reflection. The authors provided the explanation for distinguishing global (static) reasoning style control from instance-specific reflection control.

- Verbosity & reasoning depth: Concerns that FR merely increases output length were responded with the evidence that shorter outputs can also improve accuracy, particularly for reasoning-tuned models, weakening the claim that gains are driven solely by verbosity.

- Comparison Fairness: The authors added prompt-ensemble baselines inspired by the prior work, showing that text prompt diversity may underperform FR, to verify that the gains are not merely due to ensemble effects.

- Computational cost: The authors clarified that steering vector computation is lightweight compared to standard forward passes. However,  one may be concerned that the output length (token number) will be scaled up by the proposed method. The comparison regarding the token length or the actual inference time will be more beneficial to show the efficacy of the method.

- Clarity and presentation: Typos and missing references will be fixed by the authors. However, AC finds that the arrangements of figures3 & 4 are still problematic in the revised submission. The captions should be fixed accordingly.

- Novelty: While the authors clarify that FR represents a higher-level framework rather than a mere addition of the existing techniques, reviewers may still view the core mechanism as incremental relative to prior steering and ensemble methods.

- Adaptivity: The AC observed that the paper relies on post-hoc selection (e.g., Best-of-N, majority vote) rather than per-instance adaptive control. However, the sentence-level dynamic control example, which will be more flexible and effective, remains qualitative.

- Sensitivity: PRM sensitivity was acknowledged but deferred to future work. The AC would like to see how the sensitivity impacts the proposed method's performance on different models.

**Reviewer Scores:**

The initial scores were 6, 2, 6, and 4, with corresponding confidence levels of 3, 4, 3, and 3.

The reviewers who gave scores of 6 are likely unchanged: the concern regarding design rationale was addressed, and their overall assessment remains positive, with remaining suggestions framed as future work.

Reviewer v6Yx, who gave a score of 2, is likely to remain negative or slightly raise to 4, as their core objections are concerning novelty and modeling assumptions, and the rebuttal may not have adequately addressed them.

Reviewer yNjN, who gave a score of 4, is likely to maintain this rating. Concerns about motivation alignment and ensemble comparisons were addressed, but the computational cost associated with the extra tokens generated was not explicitly compared.

---

### Decision · Program_Chairs · 2026-01-26

Reject